# Application of Mass Finishing for Surface Modification of Copper Cold Sprayed Material Consolidations

**DOI:** 10.3390/ma15062054

**Published:** 2022-03-10

**Authors:** Matthew A. Gleason, Bryer C. Sousa, Kyle Tsaknopoulos, Jack A. Grubbs, Jennifer Hay, Aaron Nardi, Christopher A. Brown, Danielle L. Cote

**Affiliations:** 1Department of Mechanical & Materials Engineering, Worcester Polytechnic Institute, Worcester, MA 01609, USA; magleason@wpi.edu (M.A.G.); bcsousa@wpi.edu (B.C.S.); kltsaknopoulos@wpi.edu (K.T.); jagrubbs@wpi.edu (J.A.G.); brown@wpi.edu (C.A.B.); 2KLA Instruments, Oak Ridge, TN 37830, USA; jennifer.hay@kla.com; 3VRC Metal Systems, Rapid City, SD 57719, USA; aaron.nardi@vrcmetalsystems.com; 4Chemical Engineering Department, Worcester Polytechnic Institute, Worcester, MA 01609, USA

**Keywords:** cold spray, mass finishing, powder, additive manufacturing

## Abstract

The surface roughness of additively manufactured (AM) components can have deleterious effects on the properties of the final part, such as corrosion resistance and fatigue life. Modification of the surface finish or parts produced by AM processes, such as cold spray, through methods such as mass finishing, can help to mitigate some of these issues. In this work, the surface evolution of as-produced copper cold sprayed material consolidations was studied through mass finishing. Three different copper powders attained by different production methods and of different sizes were used as feedstock. The surface topography of the cold spray deposits was measured as a function of the mass finishing time for the three copper cold spray samples and analyzed in terms of relative area and complexity, revealing an inverse correlation relating material removal rate and hardness/strength of the cold sprayed deposits. The material removal rate was also affected by the quality of the cold spray deposition, as defined by deposition efficiency (DE). Large initial drops in relative area and complexity are also likely due to the removal of loosely bonded powders at the start of mass finishing. Based on this study, the cold spray parameters that affect the rate of mass finishing have been explored.

## 1. Introduction

When considering metal additive manufacturing (MAM) techniques that rely upon powder-based feedstock for materials processing, defects, such as those associated with porosity, are of significant concern for structural components [1]. In melt-driven and powder-based MAM, defects can emerge from poor processing parameter optimization and trapped gas porosity housed within the initial feedstock particulates [2]. Beyond the influence defects and porosity have on static mechanical properties of MAM-manufactured materials (either melt-driven or solid-state in the case of cold spray) [3,4], such as the modulus of elasticity measured under tension [5,6], the yield strength [7,8], toughness [9,10], ultimate tensile strength [11,12], the as-printed surface roughness and surface finish are also a point of concern for dynamic properties, such as high and low cycle fatigue limits [13,14].

Beyond the concerns and issues surrounding mechanical performance and structural integrity in MAM-processed components, the as-printed surface roughness, and the presence of defects (either from the use of ill-suited processing parameters or the presence of pores), one must also consider manufacturing-related problems that can arise. Poor as-printed surface quality is deleterious to controlling component tolerances, ensuring process-to-part repeatability, decreased wear resistance, tribological issues, and increased corrosion susceptibility [15].

As a solid-state materials consolidation process [16], cold spray enables structural and nonstructural repairs [17], the deposition of tunable coatings [18], the reclamation or restoration of legacy components [19], and most recently, the procurement of near-net-shape components via cold spray additive manufacturing [20]. Cold spray materials processing is achieved via the transportation of powder particles within a heated carrier reaching a de Laval nozzle and accelerated to supersonic velocities until impact is reached [21]. Reaching supersonic velocities, powder particles impact a given surface at high strain rates, such that metallurgical bonds, mechanical interlocks and bonds, and severe plastic deformation are achieved until a given near-net-shape, or consolidation geometry is reached [22]. Processing parameters that are thought to be of notable importance to the research and development, as well as materials processing and engineering communities, include carrier gas species (helium, nitrogen, or air), carrier gas pressure (high or low), carrier gas temperature, standoff distance between the nozzle and target surface, nozzle traverse speed, substrate material type, feedstock material type, nozzle shape and material (polybenzimidazole or tungsten carbide, for example), and powder feed rate [23].

Although finishing techniques for cold sprayed materials consolidations and their respective as-deposited surfaces have yet to fully benefit from a wide-ranging analysis of prospective modification methods, more traditional metal AM-produced components have undergone broader consideration within the literature [24]. For example, Tyagi et al. [25], presented findings surrounding the use of electropolishing and chemical polishing methods for addressing high surface roughness associated with laser-sintered AM 316 stainless-steel specimens in the as-printed condition. That said, for cold spray processed materials to benefit from nuanced and well-understood surface finishing post-processes, one must first come to appreciate how surface finishing techniques within metal AM are compartmentalized [26]. Distilled into four distinct strata [27], surface finishing processes for metal AM include mechanical surface modification methods, abrasive surface finishing, chemical surface modification, and electrochemical surface modification methods [28] in large part [29]. That said, the cold spray community has started to venture into addressing as-deposited surface modification needs, as highlighted in [30].

Though a degree of precedent has been established for metal AM surface modification finishing techniques and their suitability, one must quantify and qualify the functionality of various processes via a surface metrology perspective [31]. From the vantage point of surface metrology and the characterization of surface topographies via metrological means within the metal AM domain, one may also consider the work of Cabanettes et al. [32], for example. In any case, to contribute further insights into the suitability of surface finishing techniques for cold sprayed materials, the present work reports observations surrounding the utility of surface modification of copper cold sprayed specimens via mass finishing through surface topographic analysis.

## 2. Materials and Methods

### 2.1. Feedstock and Cold Spray Processing

The three feedstock powders used in this study were Cu powders of varying production types, as seen in Table 1. Cu 1’s material type was designated as “DRCU-Cu-G3H0,” wherein the manufacturing method was inert gas-atomization according to the supplier, which was Powders on Demand (POD) of Solvus Global, LLC (Worcester, MA, USA). Within the designation, DRCU is an abbreviation for the Cu feedstock material product line “Direct Repair Copper.” According to the information provided by POD, this is a high-purity Cu feedstock specifically developed for cold spray applications. That said, G3H0 is an internal abbreviation associated with powder processing. As for Cu 2, the powder was also supplied by POD and manufactured via atomization and pre-processing with the intention of reducing the oxygen content. Hence, the abbreviation OFE in the Cu feedstock material product line stands for “Oxygen-free Electronic” grade Cu. Finally, Cu 3 was produced by Kymera International (Raleigh, NC, USA) using an atomization and oxide reduction hybrid technique for powder production.

An automated VRC GEN III cold spray system (VRC Metal Systems, Box Elder, SD, USA) was used to deposit the copper powder and equipped with a polybenzimidazole (PBI) nozzle. The cold spray processing conditions can be found in Table 2. The selection of these cold spray processing parameters was based upon domain knowledge shared by cold spray processing engineers from VRC Metal Systems and Solvus Global as well as processing parameters typically used by said engineers when depositing Cu onto Al 6061 substrates via cold spray. The Al 6061-T651 substrate plates were sourced from the Peterson Steel Corporation (Worcester, MA, USA) and were 127 mm in width, 152.4 mm in length and 12.7 mm in thickness.

### 2.2. Powder Characterization

The Microtrac TurboSync system (Microtrac Retsch GmbH, Haan/Duesseldorf, Germany) was used to collect powder particle size distribution (PSD) and morphology data for the feedstock powders.

An Evo MA10 (Carl Zeiss AG, Oberkochen, Germany) scanning electron microscope (SEM) was used to obtain representative micrographs of the powder feedstocks as well as the cold spray material consolidations.

An MCT microparticulate compression testing system from Shimadzu (Kyoto, Japan) was equipped with a 50 µm diameter, diamond, flat-punch indenter probe. Force was applied to individual microparticulates at a loading rate of approximately 20 mN/s until the target load of 1.4 N was achieved. Holding for only 1 s, the indenter probe was retracted after each test and cleaned of particulate debris before continuation. Ten particles were measured for each powder feedstock studied. The MCT compression tester applies test forces through an electromagnetic loading method and records a differential transformer’s displacements. The test force generating unit houses a permanent magnet as well as a force coil, such that current flowing through the force coil generates an electromagnetic force (*F*) in proportion to the coil current (*I*), as shown in Equation (1)
(1)F=2πrnBI
*r* is the radius of the force coil, *n* is the number of coils turns, and *B* is the magnetic flux density as defined in Equation (1). Since the lowering compression rod consumes the coil current, a test force correction is automatically applied to the data. Readers are encouraged to consult the Shimadzu Micro Compression Instruction Manual for more information about how the MCT compression tester is configured and operates, including details such as automated particle surface detection and force-displacement data measuring principles.

The strength of a given particulate specimen is determined using force–displacement data at a prescribed reference ratio (displacement equal to 50% of the particulate diameter in the current study) per Equation (2)
(2)$$C=f\left(x\right)=\frac{aP}{\pi d^{2}}$$
wherein *a* is 2.48 per compliance with JIS R 1639-5, *P* is the test force in N, *d* is the particle diameter in mm, and *C* is the tensile strength at said reference ratio in MPa as defined in Equation (2). To obtain the particle ultimate tensile strength, $C=f\left(50\%\right)$ was multiplied by 1.4 based on the work of Assadi et al. [33].

### 2.3. Characterization of Cold Sprayed Consolidations

A Simplimet 4000 compression mounting system (Buehler, Lake Bluff, IL, USA) and a phenolic resin was used to mount cold sprayed consolidations in an orientation perpendicular to the build direction. Standard metallographic techniques were applied using an Ecomet 300 grinder-polisher from Buehler. Metallographic preparation of the consolidated materials was finalized after achieving a mirror finish using a 0.05 µm colloidal silica suspension as the final polishing step.

Optical micrographs were obtained using a GX71 inverted metallurgical microscope (Olympus Corporation, Shinjuku City, Tokyo, Japan).

#### 2.3.1. Flat-Punch Nanoindentation for Plasticity Evaluation

Employing a quasi-dynamic approach to loading during instrumented indentation testing using a KLA iMicro Pro nanoindenter system, a 90° diamond flat-punch tip with a flat and circular apex with a diameter of approximately 10 µm, true stress (σ) vs. strain (ε) relationships were measured by nanoindentation using the novel and emergent method brought about by Hay [34]. The method for flat-punch nanoindentation returns the yield point, true stress–strain data obtained once the point of full indenter tip-to-specimen contact is made, and the Hollomon power-law coefficients K and n of the best fit applied to the post-yield observations, wherein Equation (3) is defined as
(3)$$\sigma =K\varepsilon^{n}$$

In addition to plasticity parameters and stress–strain curve extraction, hardness was concurrently reported using the same test data.

#### 2.3.2. Application of Mass Finishing

Cold spray samples were prepared for mass finishing using a HAAS vertical mini mill (Oxnard, CA, USA) to cut the consolidations into 1 cm squares. Most of the substrate material was removed leaving the 7 mm in thickness closest to the bond interface intact on each sample. The final thickness ultimately varied based upon the resultant thickness of the deposited material. Care was taken to protect the cold sprayed consolidation surfaces and keep them in the as-sprayed condition to avoid surface degradation prior to mass finishing.

Mass finishing is an abrasive finishing process that is commonly used in industry to deburr, scale, and improve the surface roughness of parts. The process generally consists of tumbling the parts in loose abrasive media supplied with a lubricant or finishing compound which wears the part over time [35]. For this study, mass finishing was completed using a Bel Air centrifugal disk Mass Finisher FMSL8 (North Kingston, RI, USA). An R1000HDPYR01 finishing media consisting of polyester bound zircon abrasives was used for all material types. Finishing was performed at 50 rpm with intermittent addition of water for lubrication and chip removal. The finishing time was varied in increments of (0, 10, 20, 30, 50, 80 min) and surface measurements were taken after each interval before the sample was subjected to further finishing.

#### 2.3.3. Surface Topography Measurements

Surface topographies were measured using a Sensofar S neox 3D Optical Profiler (Barcelona, Spain) in confocal mode. Measurements of each sample at each time were taken using a 20× objective in a 7-by-8 grid with 10% overlap and digitally stitched to produce a 5.39 µm × 5.14 µm field of view with a sampling interval of 0.69 µm. Analyses were performed in the MountainsLab Premium 9.1 (2022) software by DigitalSurf (Besançon, France).

## 3. Results and Discussion

### 3.1. Feedstock Powder Analysis

Figure 1 presents SEM micrographs (Figure 1a–c) of the three Cu powders cold sprayed during this study. Cu 1 and Cu 2 powders are primarily spherical in shape while the powder manufacturing method associated with Cu 3 resulted in relatively irregular morphologies and dissimilar microstructures relative to gas-atomized feedstock. Building upon the work of Sousa et al. [18] and others [36,37,38,39], Cu 1 and Cu 2 powder micrographs reveal typical surface textures for gas-atomized Cu powder particles. On the other hand, Cu 3 maintained a surface texture like the spray-dried Cu powder cold sprayed in Sousa et al. [18]. Still, Cu 2 was shown to have a greater density of satellites present, which is known to influence the in-flight dynamics of particles during cold spray processing.

More to the point, satellites are commonly observed defects found within atomized particulate feedstocks for powder metallurgy processes and MAM processing [40]. During atomization, satellites develop through two primary pathways [41]. The first dominant pathway responsible for satellite defect formation during atomization occurs when fine particles rapidly solidify relative to their larger particulate counterparts and impinge onto the larger droplets undergoing less rapid solidification [42]. The second dominant mechanism underpinning satellite nucleation centers upon fine particles welded to larger powder particle surfaces during re-circulation within an atomization chamber [42]. For further information regarding satellites and satellite detection and quantification of the number of satellites present within a given powder, one may consider the work of Price et al. [43].

Figure 1d,e demonstrates the PSD of the three Cu powders showing that the average size varies between the powder where Cu 1 has a much higher D_50_ of 60.5 µm, while Cu 2 has a D_50_ of 32.9 µm and Cu 3 with a D_50_ of 24.2 µm. Cu 3 also has a significant volume percentage of powder particles under 20 µm compared to the other two powders at 36.1%, where Cu 1 and Cu 2 were 2.7% and 12.3%, respectively. The difference in powder morphology is also quantified when the width to length (W/L) ratio of the powder is compared. The two gas atomized powders, Cu 1 and Cu 2, had W/L values of 0.851 and 0.827, respectively; the oxide reduced powder, Cu 3, had a W/L value of 0.685.

Microparticulate compression testing was applied to each of the three copper feedstocks studied herein. The regions of the resultant force–displacement data shared across each powder are presented in Figure 2. The average ultimate tensile strength (UTS) for each feedstock was calculated and presented in Table 3. Powder Cu 1 had the highest UTS, followed by Cu 2 and Cu 3.

### 3.2. Cold Spray Processing Analysis

Research developments and refined understandings of the cold spray process have demonstrated notable performance sensitivities to processing parameters, powder and substrate microstructures, characteristics, and powder and substrate properties. Accordingly, cold spray processing parameters of Cu feedstocks have varied significantly across the literature [44,45]. In [46], Cu—0.1% Ag powders were nitrogen cold sprayed with an inlet gas pressure of 0.8 MPa, a nozzle outlet gas temperature of 330 °C, a stand-off distance of 30 mm, a scan speed of 100 mm/s and a cooling time of 3 s. Huang et al. [47] employed a working pressure of 4.5 MPa, a working temperature of 800 °C, a standoff distance of 30 mm, a hatch distance of 3.5 mm, a gun travel speed of 50 mm/s and nitrogen as a carrier gas. And when He has been used as a carrier gas previously, the gas pressure was set to 2.6 MPa, gas temperature was set to 437 °C, spray distance was 30 mm, powder feed rate was 3 rpm and gun speed was 100 mm/s in [48]. With such variability in mind, cold spray engineers from Solvus Global and VRC Metal Systems were consulted during this work to recommend baseline processing parameters.

Using the processing parameters given in Table 1, the gas-particle nozzle flow model detailed in [49], and a critical velocity calculator based on said framework, was employed. Said calculator unveiled the particle impact velocity, $v_{p}$, critical impact velocity, $v_{c}$, critical velocity ratio (CVR), erosion velocity, $v_{e}$, and erosion velocity ratio (EVR) associated with the parameters in Table 1, the D_50_ for each powder reported in Figure 1, and the UTS measured for each particle. From the CVR for each powder under the constant cold spray processing conditions, the recommended processing parameters were far from idealized, given that, as demonstrated by [50], a target particle impact velocity should result in a CVR that is equal to 1.3 times the critical velocity. For the present study, the cold spray parameters were held constant for the three different copper powder types. Accordingly, one ought to keep in mind the fact that the presently reported properties are dependent on processing parameters in addition to material selection. Table 4 presents the calculated metrics detailed above as well as the oxygen content in wt. % ($x_{o}$) per Sousa et al. [51] analysis for Cu feedstock, wherein as defined by Equation (4).
(4)$$v_{c}=91.74\ln\left(x_{o}\right)+729.55$$

According to these calculations, Cu 3 had the lowest oxygen content, followed by Cu 2 and Cu 1 with almost double the content of oxygen.

From the CVR values tabulated in Table 4, both Cu 1 and Cu 2 would have yielded more idealized consolidated mechanical properties and densified microstructures if the processing parameters were optimized around particle size and UTS. The cold spray processing parameters associated with Cu 3 represent a nearly optimal resultant consolidation under the processing parameters prescribed. In addition to the CVR, the UTS of the feedstock, and the oxygen content of the feedstocks are in line with the processing histories of the feedstock and resultant mechanical properties of the consolidated coatings.

These calculations also allow for determination of an estimated efficiency of the cold spray process for the three Cu systems studied herein, the deposition efficiency associated with Cu 1 was 19%, Cu 2 was 51%, and Cu 3 was 71%, which is in line with the notion that the Cu 3 cold spray parameters were more ideal for that powder, resulting in the highest deposition efficiency.

### 3.3. Microstructure and Plasticity of Cold Sprayed Consolidations

The optical micrographs associated with the three Cu consolidations are shown in Figure 3a–c. Cu 1 resulted in an as-sprayed thickness of 2.67 mm; Cu 2 resulted in an as-sprayed thickness of 6.48 mm; and Cu 3 resulted in an as-sprayed thickness of 12.85 mm. Each consolidation was performed using 20 iterations in the build direction during deposition; therefore, Cu 1–3 achieved deposition thickness per layer rates of 0.1335 mm/pass, 0.324 mm/pass, and 0.6425 mm/pass, respectively.

From the optical micrographs obtained, the lack of particle–particle bonding and mechanical interlocked particulate porosity can be observed across all three consolidations studied. The degree of deformation-induced refinement of the severely plastically deformed and deposited powder particles was qualitatively observed to be least in the Cu 2 consolidation, followed by Cu 1, and then Cu 3, respectively. Said microstructures are also consistent with the true stress–strain curves measured for each of the consolidated specimens as shown in Figure 3d. Beyond the stress–strain curves and plasticity parameters discussed next, the load–displacement data obtained were also used to calculate the hardness, that is, the resistance to plastic deformation, associated with each consolidation, as shown in Table 5. The Cu 3 consolidation resulted in the highest hardness, followed by Cu1 and Cu 2 respectively.

Flat-punch instrumented indentation testing, using a nanoindenter equipped with a high load actuator, was applied to each of the three Cu cold sprayed material consolidations achieved during this research to further understand the mechanical properties of the cold spray deposits. Each of the three consolidations was tested in the cross-sectional orientation, perpendicular to the build or spray direction. The resultant average true stress–strain curves are presented in Figure 3d. Table 6 presents the yield strength, yield strain, power-law strength coefficient and the strain hardening exponent calculated for Cu 1–3. Accordingly, the flat punch indentation results show the same trend as the hardness results in terms of the strength of the three cold spray deposits, with the highest strength from Cu 3, followed by Cu 1 and Cu 2 with the lowest.

### 3.4. Surface Evolution of Cold Sprayed Surfaces

Regarding the metrics utilized herein to capture surface evolution through surface topography and metrology characterization methods, the relative area plot denotes the apparent surface area of a surface measurement at different length scales, expressed as a ratio with that measurement’s vertically projected area. Complexity is the first derivative of the relative area plot, and thus is essentially a scale-based indicator of roughness.

The general trends in relative area and complexity for each of the materials is to be expected. Abrasion of the cold sprayed material consolidations studied herein through mass finishing resulted in a reduction of surface area (relative area) and a decrease in roughness (complexity) across nearly all length scales observed and quantified via 3D confocal microscopy and data analysis methods. In addition to Figure 4, Figure 5 presents the initial and final relative area and complexity of the surfaces associated with each of the three consolidations.

At finer length scales, the relative area of the as-sprayed Cu 2 surface was largest, followed by Cu 1 and then Cu 3. Along similar lines, the as-sprayed surface with the greatest complexity (the maximal complexity observed as a function of scale) followed the same ranking; Cu 2, followed by Cu 1, and then Cu 3.

Considering Figure 4c,d, the sharp decline in both the relative area and the complexity of the as-sprayed Cu 2 surface relative to the Cu 2 surface after 10 min of mass finishing indicates a rapid removal of fine scale features. This may correspond to the elimination of loosely bound powder particles from the surface upon initial abrasion. Although the near-surface particle–particle bond strengths were able to withstand the washing off of machining coolant fluid, the bond strengths of Cu 2 powder particles following cold spray processing were not able to withstand or accommodate abrasion-induced mechanical stresses. Consistent with this claim, SEM micrographs of the as-sprayed Cu 2 surface revealed particle–particle boundaries that appeared to be only partially interlocked or bound with one another, while the cross-sectional optical micrographs presented in Figure 3 identified large pores within Cu 2.

In relation to the mechanical properties of Cu 2 cold sprayed material consolidations, Cu 2 maintained the lowest yield strength (shown in Figure 3d and Table 6) relative to Cu 1 and Cu 3, which is consistent with the observation of insufficient particle–particle metallurgical bonding. At the same time, Cu 2 was also qualitatively shown to have the greatest density of satellites (Figure 1), suggesting that the highly satellited powder could have resulted in fine features at the surface of the consolidation, which were removed as abrasion is initialized.

Considering Figure 4 and Figure 5, peak roughness values (viewed through the lens of complexity) were observed at length scales that were consistent with the powder particle diameters measured for Cu 1–3. Figure 6 illustrates the 3D surface topography evolution across all time intervals associated with mass finishing of Cu 2 for a visual representation of the surface evolution. At the same time, Figure 6 reveals the presence of a pattern of smooth islands neighboring unfinished valleys, with the size of the smooth potions growing as a function of mass finishing time. This was a consequence of large-scale surface peaks preventing access of the comparatively large particulate abrasive to the valley regions. It is to be expected then that wear would take place primarily at these peak regions resulting in the smooth-to-unfinished pattern that is seen. To produce an even surface finish sufficient material must be removed to eliminate these large-scale features. These large-scale features are common to unfinished cold spray surfaces, and, as such, one should select and appropriately aggressive media and longer finishing times for such surfaces.

Interestingly, the relative magnitudes in the drop in relative area and complexity appear more similar for Cu 2 and Cu 3 than either with Cu 1, despite them having a greater variation in yield strength and hardness (Cu 2 is lowest, Cu 3 is highest). This is counterintuitive as one would expect an inverse relationship between yield strength and hardness and surface smoothing rate. This may point to different failure modes such as individual particles flaking off the surfaces due to low adhesion bond strength with neighboring particulates. Cu 1 had the lowest deposition efficiency and CVR which are indicators of poor bonding. Additionally, the Cu 1 powder had the highest ultimate tensile strength suggesting that it required more kinetic particle impact energy to metallurgically bond. A poorly bonded particle–particle structure within a given cold sprayed material consolidation would be more prone to abrasion.

Regarding Cu 1 and Cu 2 surfaces, the roughness was found to increase at the finest length scales once the initial mass finishing time step was applied. Considering the surface features present across the powder particles presented in Figure 1 for Cu 1 and Cu 2, the initial increase in roughness at the finest length scales can be attributed to the abrasive media imparting additional roughness at fine length scales as the surface finishing process began to remove surface material and features as the as-sprayed surfaces were modified. As shown in Figure 7, the surface features found along Cu 1 and Cu 2 particles were initially conserved following deposition processing via cold spray, which is consistent with the abrasive media imparting fine-scale roughness as mass finishing began.

Although material removal rate (MRR) was not directly measured the rate of surface smoothing as denoted by the progression of relative area and complexity can be taken as an indicator. From Figure 4a–f, the mass finishing-induced material removal rate, after 10 min was Cu 2, followed by Cu 1 and then Cu 3. As for mass finishing times ranging between 10 to 80 min, Cu 2 no longer maintained the greatest rate of material removal; rather, Cu 1 surfaces experienced a decrease in the relative area as a function of scale, followed by Cu 3 and then Cu 2. Figure 4e,f demonstrates the fact that the Cu 3 surface maintained the steadiest rates of material removal as a function of time, followed by Cu 2 and then Cu 1.

The hardness of the consolidations can be linked to the MRR to contextualize the results of surface evolution in terms of abrasive wear theory according to the work of Cariapa et al. [52], wherein they considered such a theoretical framework in the context of mass finishing. From this work, Equation (5) presents the following relation between MRR and the hardness (i.e., the mean contact pressure, which is an indicative metric related to material resistance to plastic deformation), such that
(5)$$MRR=\alpha e^{P_{r}}+\beta H_{r}^{\gamma }=C_{1}+\beta {\left(\frac{H_{Cu}}{H_{media}}\right)}^{\gamma }\;=C_{1}+\left(\frac{\beta }{H_{media}^{\gamma }}\right)H_{Cu}^{\gamma }=C_{1}+C_{2}H_{Cu}^{\gamma }$$
wherein α, $P_{r}$, β, γ, $H_{r}$, $C_{1}$, $H_{Cu}$, $H_{media}$, and $C_{2}$ represent a fitting parameter, the density ratio of the consolidation and the media, a fitting parameter, another fitting parameter, the hardness ratio of the consolidation and the media, a constant, the hardness of the Cu consolidations, the hardness of the media, and a final constant, respectively, as defined by Equation (5). Since the mass finishing conditions and media were held constant during the present work, one may recast MRR as an inverse proportionality relative to the hardness of the Cu consolidations in Equation (6)
(6)$$MRR\propto \frac{1}{H_{Cu}}$$
which was consistent with the findings presented herein (note that the fitting parameter γ is negative). Given this relationship, future investigation can be completed regarding control of the feedstock powder, cold spray parameters and mass finishing parameters to further understand how to optimize surface finishing for cold spray deposits.

## 4. Conclusions

Recent literature has shown the importance of a component’s surface finish after AM processing, such as with cold spray, as surface roughness can affect the final component’s corrosion resistance, fatigue life, and performance in functional applications, including antimicrobial surfaces. In this study, post-cold spray mass finishing was explored as a means of mitigating detrimental surface roughness effects on as-sprayed copper deposits. The copper samples’ surface topographies were found to be linked to several material- and process-dependent properties, such as powder particle size, shape, oxygen content, and manufacturing method, as well as cold spray processing conditions. The topographical characteristics also can be associated with the material’s wear rate, as measured after mass finishing through several hardness and strength metrics. More specifically:An inverse relationship between material removal rate and hardness was seen.Material removal rate was also affected by the quality of cold spray coating and densification of powder during processing.
Quantitatively, a lower DE can lead to an increase in wear removal rate.
The large immediate drop in relative area and complexity was likely due to removal of loosely bonded powders at the beginning of the finishing process.Peaks and valleys in the initial cold spray surfaces cause irregular wear removal at lower times but would even out with optimization of mass finishing time.The finishing times in this study were long in comparison to what is usually used in industry since high initial roughness at large scales require a large amount of material to be removed to achieve appropriate smoothness.
We would recommend harder, entirely ceramic media for proper finishing in the future.


Based on this study’s findings, mass finishing has shown to be a suitable surface modification technique for as-sprayed copper samples, particularly as a replacement for conventional machining techniques.

## Figures and Tables

**Figure 1 materials-15-02054-f001:**
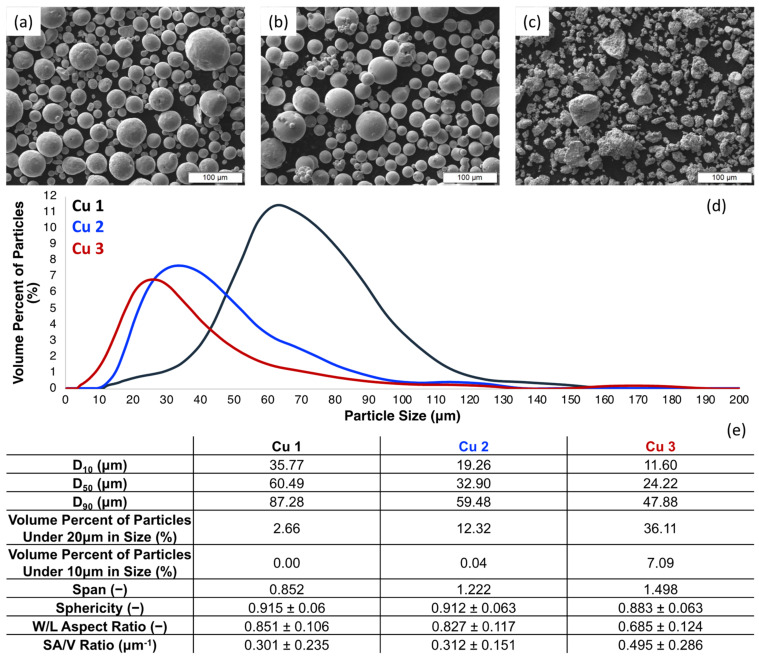
(**a**–**c**) SEM micrographs, (**d**) particle size distributions, (**e**) tabulated particle size and shape data for Cu 1–3.

**Figure 2 materials-15-02054-f002:**
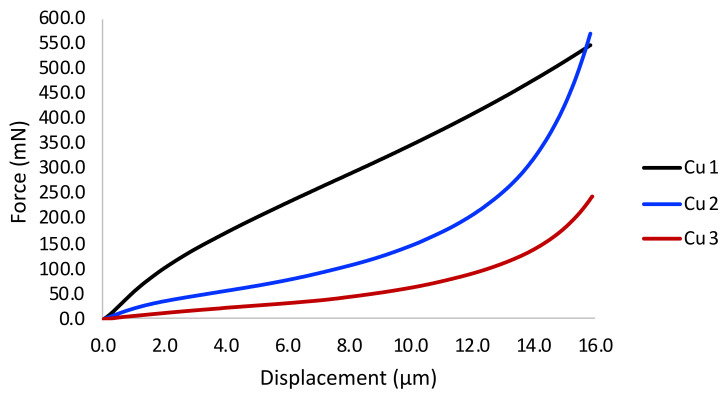
Presentation of the average force vs. displacement curves obtained from microparticulate compression testing of the three copper powders.

**Figure 3 materials-15-02054-f003:**
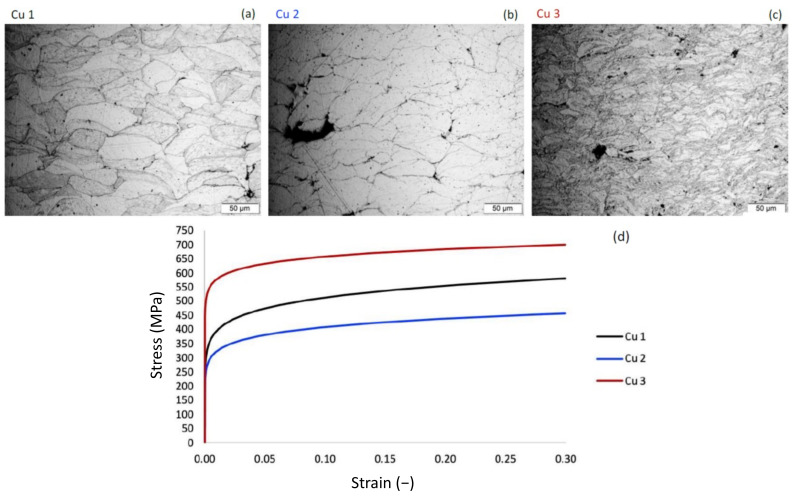
(**a**–**c**). Optical micrographs of the cross-sectional view of the cold sprayed samples created from powders Cu 1–3. (**d**). Presentation of the average true stress vs. true strain curves determined via flat-punch indentation testing.

**Figure 4 materials-15-02054-f004:**
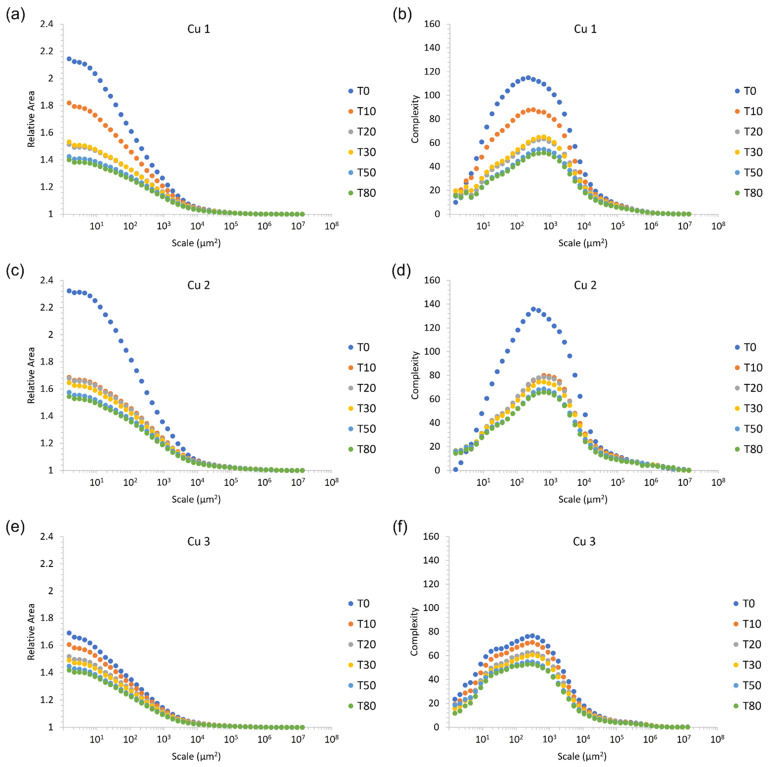
Surface evolution as a function of finishing time for each of the 3 Cu powders. (**a**,**c**,**e**). Relative area as a function of finishing time, (**b**,**d**,**f**). complexity as a function of finishing time.

**Figure 5 materials-15-02054-f005:**
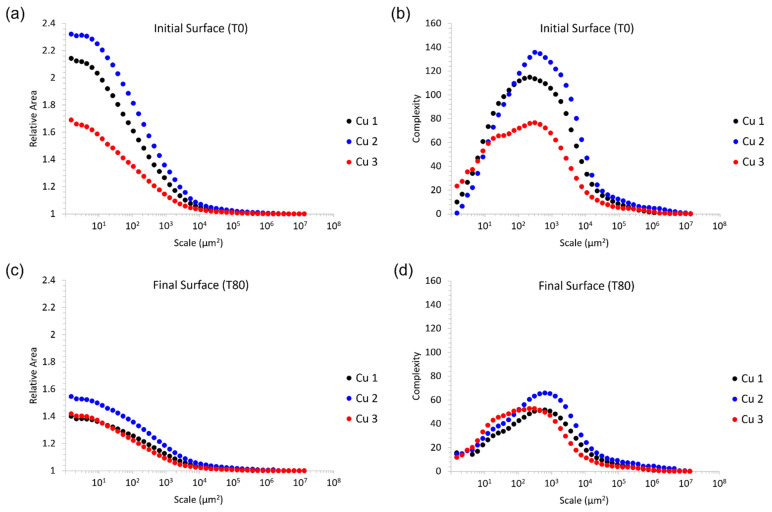
Initial vs. final surface finish. (**a**,**c**) Relative area compared for Cu 1–3, (**b**,**d**). Complexity compared for Cu 1–3.

**Figure 6 materials-15-02054-f006:**
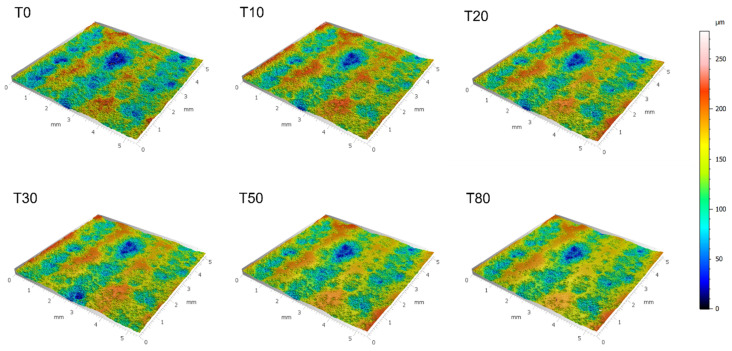
3D confocal surface topography of the Cu 3 surface as a function of mass finishing time.

**Figure 7 materials-15-02054-f007:**
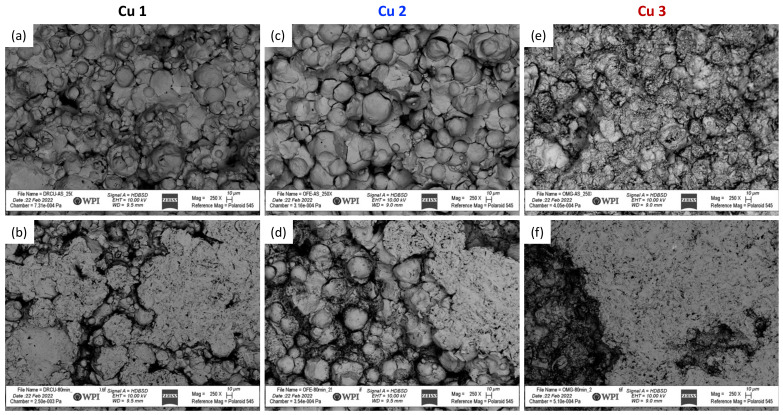
(**a**,**c**,**e**) present SEM micrographs of the as-sprayed Cu 1, Cu 2, and Cu 3 cold sprayed material consolidations, respectively. Similarly, (**b**,**d**,**f**) present SEM micrographs of the Cu 1, Cu 2, and Cu 3 cold sprayed material consolidations, respectively, following the application of mass finishing for 80 min. Note that (**b**,**d**,**f**) were captured from regions of the modified surfaces that were along the interfacial or transitional zones between the smooth islands and rough valleys previously discussed.

**Table 1 materials-15-02054-t001:** Feedstock powder material details.

Material ID	Material Type	Manufacturing Method
Cu 1	DRCU-Cu-G3H0	Gas-atomization
Cu 2	OFE-Cu	Gas-atomization + Pre-processing
Cu 3	Oxide Reduced Copper	Gas-atomization + Oxide Reduction

**Table 2 materials-15-02054-t002:** Cold Spray processing parameters.

Nozzle	Gas Temperature	Carrier Gas	Gas Pressure	Spray Angle	Standoff Distance	Traverse Speed	Feeder Rate	Substrate
0071, PBI	425 °C	N_2_	2.6 MPa	90 °	25.4 mm	120 mm/s	4 RPM	Al 6061-T651

**Table 3 materials-15-02054-t003:** Ultimate tensile strength, average force required to reach a displacement equal to half the particle diameter, average displacement, and average powder particle diameter tested, for Cu 1–3.

Sample	Ultimate Tensile Strength (MPa)	Force to Achieve 50% Compression (mN)	Average Displacement (µm)	Average Powder Particle Diameter (µm)
Cu 1	437.61 ± 69.05	958.34 ± 292.31	24.67 ± 4.94	49.32 ± 9.88
Cu 2	368.10 ± 30.34	237.69 ± 154.60	12.84 ± 3.77	25.65 ± 7.55
Cu 3	212.95 ± 95.74	172.15 ± 104.02	14.51± 2.17	28.99 ± 4.36

**Table 4 materials-15-02054-t004:** Cold spray processing parameter-based critical velocity calculator outputs and analytically identified oxygen content.

Powder	Impact Velocity (m/s)	Critical Impact Velocity (m/s)	Critical Velocity Ratio	Erosion Impact Velocity (m/s)	Erosion Velocity Ratio	Oxygen Content (wt. %)
Cu 1	45	603	0.762	1205	0.381	0.2517
Cu 2	623	592	1.053	1184	0.520	0.2233
Cu 3	726	515	1.408	1031	0.704	0.0965

**Table 5 materials-15-02054-t005:** Flat-punch indentation hardness, the applied load, and the indentation depth for Cu 1–3.

Sample	Hardness (GPa)	Applied Load (mN)	Indentation Depth (nm)
Cu 1	2.36 ± 0.32	268.3 ± 23.3	2046 ± 3
Cu 2	1.71 ± 0.14	214.1 ± 6.9	2052 ± 4
Cu 3	3.16 ± 0.19	324.9 ± 3.6	2041 ± 1

**Table 6 materials-15-02054-t006:** Flat-punch indentation yield strength, yield strain, strength coefficient, and strain hardening exponent for Cu 1–3.

Sample	Yield Strength (MPa)	Yield Strain	Strength Coefficient (MPa)	Strain Hardening Exponent
Cu 1	348.8 ± 71.2	0.0030 ± 0.0006	663.9 ± 54.7	0.112 ± 0.031
Cu 2	280.7 ± 15.6	0.0024 ± 0.0001	517.3 ± 32.4	0.101 ± 0.017
Cu 3	555.9 ± 52.6	0.0048 ± 0.0005	748.7 ± 25.0	0.056 ± 0.022

## Data Availability

All the data is available within the manuscript.

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
