# Peer review of "Application of Mass Finishing for Surface Modification of Copper Cold Sprayed Material Consolidations"

_materials, 2022, doi:10.3390/ma15062054_

Round 1
Reviewer 1 Report
It is hard to understand how the term ‘Mass finishing’, according to https://doi.org/10.1016/S0026-0576(01)85268-5, is connected to the manuscript.
In Table 2, cold spraying parameters should be justified.
In Table 2 and in the text, all parameters should be given in international SI units, but no in psi, bar, etc.
Line 103: 1400 mN may be equal to 1.4 N?
Lines 178-185, 252: Rounding of the values of the same parameters should be unified.
Lines 193-204, 266-257, 283-98: Too many units in such expressions as ‘437.61 MPa ± 69.05 MPa’, …, ‘28.99 μm ± 4.36 μm’.
The Kelvin/Celsius and ‘deg.’/° temperature notation should be unified throughout the manuscript.
The lines on the graphs should be drawn in different types so that they can be distinguished after black and white printing. The same is for Figures 4 and 5, but for the dot types.
Equations should be numbered and referenced in the text.
The conclusions are too abstract and cloudy. It should clearly describe, point by point, what results have been achieved and what follows from them.
Reviewer 2 Report
- Table 1 and sub-chapter 2.1. More details about different manufacturing methods of copper powders should be given (Table 1). The explanations of all used abbreviations in this table are necessary. The manufacturing method cannot be described as Proprietary of ...
- Lines 100-101: `Shimadzu (Kyoto, Kyoto, Japan)`. Twice a Kyoto, an error.
- What was a substrate material?
- Line 149: `7mm`. A space between a value and a unit is obligatory.
- Line 175: `spray-dried Cu powder cold sprayed in Sousa et al.`. Please give again the reference number.
- Line 176: `satellites present,`. What satellites?
- Line 215: `seconds.`, use a proper SI unit symbol [s].
- Line 230: `target particle impact velocity should result in a CVR that is equal to 1.3 times the critical velocity.`. ... result in a critical velocity rate that (rather: which) is equal to 1.3 times the critical velocity ... Something is wrong, CVR is higher than CVR?
- Lines 233-234: `processing parameter dependent in addition to material dependent.` dependent to (rather: on) dependent? Correct this phrase.
- Table 3: the units should be given in the table header rather than in repeating them in columns (m/s, wt.%).
- Lines 267-274. Some values are presented with 7 significant figures (numbers) such as e.g. 268.3252, at high standard deviation, in this example 23.3078. This is unreasonable and I suggest limiting the number of significant figures to no more than four (268.3 and 23.3, respectively).
- Lines 267-274. The depth is shown by values with decimals so that at the precision of 0.1 nm. Has such precision been obtained? If not, please give no decimals; e.g., for a first depth value and an SD, as 2045 and 3 nm, respectively.
- Line 278-279: `Each of the three consolidations were tested`. Rather: Each ... was tested.
- What do the authors mean by Cu consolidations? Define it early in the text.
- Figures 4 and 5: what is a complexity shown as an output parameter in several graphs? There is no such definition, and line 306 gives not clear image in this aspect.
- Finally, the discussion in subchapters 3.2 and 3.3 must be developed. Now, there is no discussion of data included in Figs. 3 to 6, only a presentation of results.
- Lines 326-347: if I understood well, the material removal rate increases with hardness? It is against all known to me experimental data. Besides, this paragraph is not supported by aby here presented results. Concluding, this paragraph should be removed from an article as neither wear rate nor hardness has been investigated in this research.
All these remarks have been indicated by color in the attached review.

Reviewer 3 Report
The present work is devoted to the study mass finishing for surface modification of copper cold spray coatings. The work is interesting but some major revision are required before publication.
-Please rewrite the abstract section giving also quantitative data on the results and with a more comprehensive discussion
-Please improve the introduction section adding and discussing recent works where coatings where applied to modify the roughness of AM parts such as “L. Pezzato, M. Dabalà, Silvia Gross, K. Brunelli, Effect of microstructure and porosity of AlSi10Mg alloy produced by selective laser melting on the corrosion properties of plasma electrolytic oxidation coatings, Surface and Coatings Technology, Volume 404, 2020, 126477, https://doi.org/10.1016/j.surfcoat.2020.126477.”
-Considering that is the core of the work I suggest to add a proper scheme of the mass finishing system
-Please improve the characterization of the cross section of the coatings, in order to understand also the total thickness of the coating
-Please improve the characterization of the surfaces of the samples adding OM and SEM images before and after the mass finishing process
-Considering the mechanical properties, I suggest to add proper tables instead of writing them in the manuscript
-Considering Fig.3 the curves are the average of various measurements? Please specify
-The main lack of the manuscript, together with lack of characterization, is the total absence of proper discussion of the results. What is the mechanism that generate the differences between the samples? How are the results in comparison with the literature etc…Now the work is more an experimental report than a scientific paper and proper discussion is mandatory
-Please rewrite the conclusion section as bullet points
Reviewer 4 Report
After reviewing the manuscript. Please find below comments and suggestions to be included in the manuscript.
Line 71 to 73. In the sentence “That said, the cold spray community has started to begin to venture into the realm of addressing as-deposited surface modification needs, as highlighted in [29]” please remove “to begin” and perhaps replace the term “realm” for other.
Line 137-139. There are many published works in the literature that try to deal with this topic, which is complex in nature. Usually, inverse algorithms are promising to determine the stress-strain curve from nanoindentation. Nonetheless, there are several assumptions and usually depend on each material. In this case, why did the authors decide to use a single patent in this topic instead of peer review literature or a more widely accepted protocol in the literature?
See for instance: Pathak, Siddhartha and Surya R. Kalidindi, Spherical nanoindentation stress-strain curves. Materials Science and Engineering: R: Reports, 2015. 91: p. 1-36.
Line 141. The sentence “Holloman power-law” should be “Hollomon power-law”
Line 165. In the sentence “digitally stitched to produce a 5.39 x 5.14 field of view will a sampling interval of 0.69 µm.” it should be “with” instead of “will”
Figure 1d. A finer scale is desired, i.e., more numbers rather than only 10 and 100 um in the x-axis. Please use the same significant figures for the values reported in that table in order to be consistent.
Line 213-214. It is more widely used the convention 330°C. Please replace in all the manuscript.
Line 276. In the sentence “Flat-punch nanoindentation or instrumented indentation” these are not synonyms that can be used interchangeably. Please correct.
Line 305-307. For the readership's sake, Please provide context and definition about the concept of "complexity" in the manuscript.
Line 335, the relationship drawn should be discussed in the light of the results obtained using internal cross-references. For instance:
[According to the equation "MRS->H" it can be observed that as the hardness increase (see section 3.3) the MRS increases as well (see Figure 5 xx)]
This should be expanded and provide further discussion since this is one of the main important conclusions from this work and might help other researchers in the field.
Round 2
Reviewer 2 Report
I have no other remarks and appreciate a lof of work performed to improve the article and delete some errors and unclearities.
Reviewer 3 Report
Considering that the authors have properly ansered to all the main issues of the first revision i suggest publication in its form for this work